# Electronic clinical decision support tool for assessing stomach symptoms in primary care (ECASS): a feasibility study

Greg Rubin ![ORCID],[1] Fiona M Walter ![ORCID],[2] Jon Emery ![ORCID],[3] Willie Hamilton ![ORCID],[4] Zoe Hoare ![ORCID],[5] Jenny Howse,[6] Catherine Nixon,[6] Tushar Srivastava,[7] Chloe Thomas ![ORCID],[7] Obioha C Ukoumunne ![ORCID],[8] Juliet A Usher-Smith ![ORCID],[2] Sophie Whyte,[7] Richard D Neal ![ORCID][9]

For numbered affiliations see end of article.

**Correspondence to**
Professor Greg Rubin;
Gregory.Rubin@newcastle.ac.uk

## ABSTRACT

**Objective** To determine the feasibility of a definitive trial in primary care of electronic clinical decision support (eCDS) for possible oesophago-gastric (O-G) cancer.

**Design and setting** Feasibility study in 42 general practices in two regions of England, cluster randomised controlled trial design without blinding, nested qualitative and health economic evaluation.

**Participants** Patients aged 55 years or older, presenting to their general practitioner (GP) with symptoms associated with O-G cancer. 530 patients (mean age 68 years, 58% female) participated.

**Intervention** Practices randomised 1:1 to usual care (control) or to receive a previously piloted eCDS tool for suspected cancer (intervention), for use at the discretion of the GPs, supported by a theory-based implementation package and ongoing support. We conducted semistructured interviews with GPs in intervention practices. Recruitment lasted 22 months.

**Outcomes** Patient participation rate, use of eCDS, referrals and route to diagnosis, O-G cancer diagnoses; acceptability to GPs; cost-effectiveness. Participants followed up 6 months after index encounter.

**Results** From control and intervention practices, we screened 3841 and 1303 patients, respectively; 1189 and 434 were eligible, 392 and 138 consented to participate. Ten patients (1.9%) had O-G cancer. eCDS was used eight times in total by five unique users. GPs experienced interoperability problems between the eCDS tool and their clinical system and also found it did not fit with their workflow. Unexpected restrictions on software installation caused major problems with implementation.

**Conclusions** The conduct of this study was hampered by technical limitations not evident during an earlier pilot of the eCDS tool, and by regulatory controls on software installation introduced by primary care trusts early in the study. This eCDS tool needed to integrate better with clinical workflow; even then, its use for suspected cancer may be infrequent. Any definitive trial of eCDS for cancer diagnosis should only proceed after addressing these constraints.

**Trial registration number** ISRCTN125595588.

## Strengths and limitations of this study

► This feasibility study used an electronic clinical decision support (eCDS) tool for possible oesophago-gastric cancer that had been previously developed and piloted by Macmillan Cancer Support in collaboration with a provider of GP clinical software (TPP).

► This was a pragmatic study in primary care, with general practitioners (GPs) using the eCDS tool at their discretion.

► Implementation of the intervention with GPs was theory-based, using educational outreach, with ongoing clinical and technical support provided by the research team.

► Participation in the study was significantly hampered by technical problems relating to the interface between the eCDS tool and the GP clinical system that had not been reported in an earlier pilot, and by restrictions on installation of software on GP systems introduced without warning by some primary care trusts during the implementation phase of the study.

► For some GPs in the intervention arm of the study, the release of updated National Institute for Health and Care Excellence guidance on management and referral of suspected cancer superseded the need to use a decision support tool.

## BACKGROUND

Recognising the significance of symptoms that may indicate an underlying cancer is fundamental to clinical practice in primary care. However, many patients in primary care present with low-risk symptoms, and even 'red flag' symptoms have a lower positive predictive value compared with patients seen in specialist care.[1] Research using data from primary care populations has generated robust estimates of the risk of cancer in symptomatic patients presenting to general practitioners (GPs),[2 3] from which risk assessment

tools have been developed[4][5] and then evaluated.[6] In the UK, these tools have also been transformed into electronic clinical decision support (eCDS) formats.[7] Their implementation has been promoted by the report of the Independent Cancer Taskforce for England in 2015,[8] though they remain an underused resource.[9]

Nevertheless, uncertainty exists about the effectiveness of clinical decision support (CDS) for potential cancer symptoms and how to best incorporate it into clinical practice. One systematic review identified the features critical to the success of CDS interventions.[10] A second review of eCDS tools found that they improved practitioner performance in 64% of the 97 included studies,[11] while a third identified prompt fatigue as a strong reason for failure of eCDS.[12] Most recently, a systematic review of CDS to support cancer diagnosis in primary care identified nine studies (four randomised controlled trials (RCTs) and concluded that the optimal mode of delivery remains unclear.[13] However, an early study of CDS for suspected cancer found that it was more likely to be embedded in clinical practice if it supported rather than superseded clinical judgment.[14] We therefore undertook a study of the feasibility of a trial of an eCDS tool for suspected cancer. The earlier development of this eCDS had been led by Macmillan Cancer Support. We used oesophago-gastric (O-G) cancer as our exemplar site.[15] We aimed to optimise an intervention based on this eCDS tool, establish its acceptability and collect relevant data to inform the design of a subsequent definitive trial. We also sought to generate new knowledge on the processes of eCDS in primary care and to obtain preliminary evidence on the effectiveness, implementation and cost-effectiveness of eCDS.

## METHODS

This was a multisite feasibility study using a cluster RCT design without blinding, supported by the North Wales Organisation for Randomised Trials in Health Clinical Trials Unit. We used a version of the Macmillan eCDS tool based on the Hamilton risk assessment tools,[2] for the purpose of the study limiting its use to symptoms of possible O-G cancer. The tool had been developed by Macmillan Cancer Support with TPP (SystmOne) and BMJ Informatica and had been distributed in 2013 as a National Awareness and Early Diagnosis Initiative (NAEDI) project to 439 practices in 15 cancer networks for a pilot period of 9 months.[15] It provided a drop-down box with an interactive risk calculator which could be opened at the GP's discretion. Additional symptoms could be entered by the GP and a value could be generated for the risk of a currently undiagnosed O-G cancer.

The protocol for this study has been previously published.[16] In brief, patients aged 55 years and older, presenting to their GP with symptoms associated with O-G cancer[2] and capable of informed consent were recruited from general practices in the North East and North Cumbria and the Eastern Local Clinical Research Networks. An automated record search tool to identify eligible patients for the study was developed in collaboration with Information and Computing Services at Stockton-on-Tees Primary Care Trust (PCT). This was tested and retested to maximise its sensitivity, prior to being supplied to participating practices and run on a weekly basis. Eligible patients received by post from their GP an information pack comprising an invitation letter and participant information sheet, together with a consent form to permit access to their primary and secondary care records for follow-up data. This form was returned by post to the research team. Practices (clusters) were randomised by North Wales Organisation for Randomised Trials in Health Clinical Trials Unit to receive the eCDS tool or to usual care, stratified by the region in which they were located. Allocation was balanced within region, randomising practices on a 1:1 ratio using block sizes of 2. Practices were randomised in pairs to maintain allocation concealment.

### Implementation of the eCDS tool

Intervention practices received an implementation package based on principles of educational outreach.[17] This comprised an initial meeting of 30–60 min on practice premises between a GP from the research team (FMW or GR) and the practice clinicians. The meeting included a presentation on the development of the eCDS tool, the way that it interfaced with their clinical system, how it related to NICE guidance on referral for suspected cancer, when and how to use the tool and how to interpret the results. The practice manager for each practice was visited by a member of the research team to support the uploading of the eCDS software and to explain the processes for patient searches. The research team provided technical support throughout the study. GPs had access as necessary to peer-to-peer support from clinicians in the research team and received study newsletters throughout the study. All practices received free access to the Royal College of General Practitioners online learning module on cancer diagnosis.

The study was limited to practices operating the TPP (SystmOne) clinical system. Practices that had previously participated in the NAEDI eCDS initiative were excluded. This was a pragmatic study, meaning that the eCDS tool could be accessed and the output used at the GP's discretion. As configured for this study, it did not generate automatic 'prompts'.

The installation of software on practice computer systems for the purpose of research became subject to new regulatory controls early in the study. The implementation of these controls differed between PCTs, but the way they were applied in the northeastern PCTs resulted in long delays in installation of the eCDS software, disrupting the timely activation of the intervention arm of the study.

### Process and outcome measures

Service-related outcome measures were referral rate by referral pathway in each arm of the study, conversion

(proportion of referrals with a cancer diagnosis) and detection (proportion of OG cancer detected through a 2-week wait referral) rates. We also sought estimates of recruitment and consent rates among those eligible for inclusion. Practitioner-related outcomes were frequency of use of the eCDS and attitudes to, and role of, the tool in clinical practice.

## Data collection

BMJ Informatica supplied a version of the tool modified to our specification to enable capture of data related to its use (symptoms entered and risk score generated) on the practice computer network but separate from the GP clinical system and not visible to users. In addition to these data, research staff collected individual patient data from GP records (online supplemental table 1) 6 months after the index consultation, using a previously developed data extraction template. Where necessary, hospital gastroenterology units were visited to retrieve data on secondary care procedures and diagnoses.

Semistructured 1:1 interviews with GPs in intervention practices were conducted to identify and gain an understanding of the facilitators and constraints influencing implementation of eCDS in routine practice.

## Sample size and data analysis

The study was designed to provide sufficient process data and enough participants with O-G cancer to provide estimates of patient participation rate, use of eCDS and overall percentages for binary outcomes. We aimed to recruit a minimum of 40 practices with 1:1 randomisation between intervention and control arms. Estimates of sample size were based on data from the Office of National Statistics, Trent Cancer Registry, previous experience of recruitment to primary care trials and pilot searches of primary care records, and are fully stated in our protocol paper.[16] We anticipated that over a 16-month period, 2000 eligible patients would be asked to participate; 1600 patients would be recruited (800 in each arm) and 64 of these (32 in each arm) would have O-G cancer. The target sample size was decided based on estimating feasibility parameters and providing a sufficient amount of process data. For example, if the consent rate is 80%, 2000 eligible participants are large enough to estimate this with a 95% CI of 78% to 82%, and 64 participants with O-G cancer are large enough to estimate percentages for binary outcomes with 95% CIs no wider than 37%–63% overall and no wider than 32%–68% within each arm.

Characteristics of the practices and participating patients were summarised using numbers and percentages for categorical variables and means and ranges for quantitative variables. Logistic regression was used to compare the study arms with respect to referral pathways used, use of gastroscopy and cancer diagnoses in crude (unadjusted) analyses and analyses adjusted for region and practice size. No p values were reported as this was a feasibility study.

## Health economic methods

An economic model was developed in Microsoft Excel 2007 to evaluate the cost-effectiveness of using the eCDS tool in patients presenting to the GP with symptoms potentially representing O-G cancer. The model was informed by a conceptual mapping exercise, study data and published literature obtained via rapid literature reviews. Incremental outcomes were modelled using probabilistic sensitivity analysis to enable uncertainty to be estimated. Detailed costings for installation and training were not available. Therefore, a maximum justifiable cost analysis was carried out to estimate what the maximum cost of eCDS installation and training could be while still allowing it to be cost-effective. A comprehensive description of these methods is available.[18]

## Patient and public involvement

A patient reviewed the research proposal prior to submission for funding and commented on the documents included in the patient recruitment pack. Patients also participated in the independent study steering committee.

## RESULTS

We recruited 42 practices to the study, 21 randomised to each arm. Eight practices withdrew over the time course (seven intervention and one control). The total recruitment period was from November 2015 to the planned end date of December 2017. However, practices commenced patient recruitment as their software was installed and induction was completed. Therefore, over a median patient recruitment period of 17.5 months (range 9–22 months), we recruited 530 patients in total (table 1 and figure 1). Two-thirds (68%) of patients identified through weekly searches of the clinical and prescribing records of participating practices proved ineligible on scrutiny of the clinical records. The most frequent reason was incorrect identification by a prescription 'flag', most commonly triggered by prescription of acid-suppressing drugs for gastric cytoprotection or reauthorisation of long-term medication.

The baseline characteristics of participants in control and intervention practices were comparable (table 2). Practices in each arm were of comparable mean size; the mean number of full-time equivalent GPs in each practice was not available.

The number of patients recruited was considerably greater in the control arm. This was due to the unforeseen delays, previously referred to, at the point of installation of the eCDS tool in a number of intervention practices.

The eCDS tool was used on eight unique patients by five GPs in five intervention practices over the course of the recruitment period. Usage data for three practices were lost because the software was removed without prior discussion with the research team. No adverse events were reported.

Estimates of the intervention effect on the referral pathways used, use of gastroscopy and cancer diagnoses are reported in table 3.

**Table 1** Participant recruitment

|  | Total | Intervention | Control |
|---|---|---|---|
| Patients identified as potentially eligible by clinical record searching | 5144 | 1303 | 3841 |
| Patients invited (following GP screening of searches for ineligible patients) | 1623 | 434 | 1189 |
| Patients consenting to study | 530 | 138 | 392 |
| Patients with complete follow-up data | 527 | 137 | 390 |
| Patients with incomplete or no follow-up data | 3 | 1 | 2 |
| Patients recruited as % of those potentially eligible | 10.3% | 10.6% | 10.2% |
| Patients recruited as % of those invited | 32.7% | 31.8% | 33.0% |

GP, general practitioner.

## Qualitative findings

Nine GPs were interviewed across six practices enrolled in the intervention arm (two practices from the northeast and four from the eastern area, table 4).

Five of the nine GPs interviewed were female. Five had been registered for >20 years. Their practices had a spread of patient list sizes and included a small urban practice (n<4999) and two large rural practices (n>10 000). GPs were interviewed at variable time points after their first induction into using the tool; the mean interval between induction and first interview was 11

months (range 2–19 months). Four of the nine GPs (three female and one male) were interviewed more than once in order to see if their views of eCDS changed over time.

Use of eCDS by GPs in participating practices, as identified by computer records, was very low and only loosely consistent with use claimed during interviews. Problems with its use were identified by all GPs interviewed. These related to both access and use of the tool, and integrating the tool within clinical practice (table 5). The most common challenges with access and

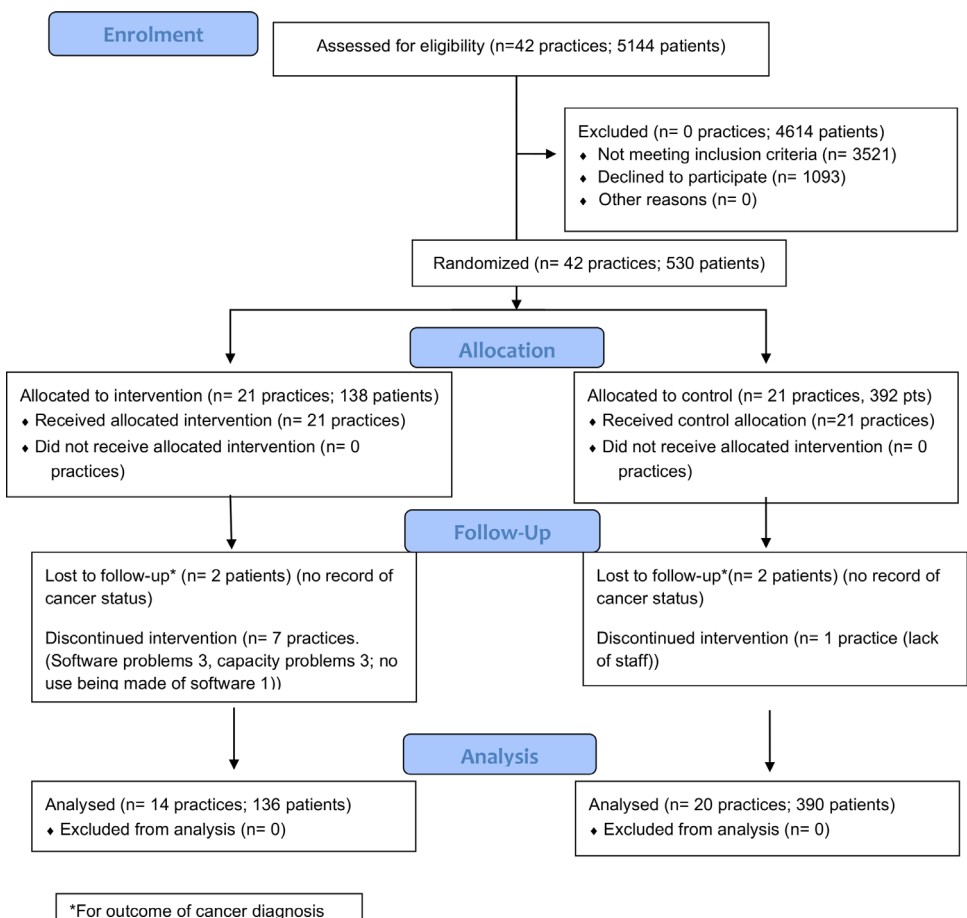

**Figure 1** CONSORT 2010 flow diagram ECASS. CONSORT, Consolidated Standards of Reporting Trials; ECASS, Evaluation of a Computer Aid for Assessing Stomach Symptoms.

**Table 2** Baseline characteristics of practices and participating patients

| Characteristic | Intervention | Control |
|---|---|---|
| Patients | N=138 | N=392 |
| Female, n (%) | 84 (60.9) | 225 (57.4) |
| Age (years), mean (SD) | 68.4 (8.7) | 68.0 (8.6) |
| Region, n (%) | | |
| North East | 27 (19.6) | 227 (57.9) |
| Eastern | 111 (80.4) | 165 (42.1) |
| Practices | N=21 | N=21 |
| Region | | |
| North East | 11 | 10 |
| Eastern | 10 | 11 |
| List size, mean (range) | 9682 (1686–15 447) | 10 161 (2371–19 934) |

use were 'lack of integration of the software with the clinical systems' (n=7) and 'slow to access and/or use' (n=6).

When speaking about integrating the tool within clinical practice, GPs were frustrated with the apparent mismatch between the tool and the clinical context in which they practised, where codes were often not used and time was always a constraining factor on what could be completed. Several had concerns about the accuracy of the data used in the tool. Two GPs additionally commented on how the tool was not yet embedded in their clinical practice and how the new NICE cancer guidelines superseded the tool in terms of decision support.

> I think the benefits of the other tools are clearer, just because of the experience we've got with them and because they're accepted by QOF and the local CCG, that sort of thing. (GP1)

**Table 3** Comparison of outcomes between trial arms

| Outcome | Intervention % (n/N) | Control % (n/N) | Crude comparison OR | 95% CI | Adjusted comparison OR | 95% CI |
|---|---|---|---|---|---|---|
| Patient was referred. | 51.1 (70/137) | 48.6 (189/389) | 1.11 | 0.70 to 1.75 | 1.13 | 0.73 to 1.76 |
| Patient was referred via standard or 2WW pathway. | 48.9 (67/137) | 45.0 (172/382) | 1.17 | 0.73 to 1.88 | 1.17 | 0.75 to 1.83 |
| Patient was referred via standard pathway. | 26.3 (36/137) | 19.1 (73/382) | 1.51 | 0.90 to 2.53 | 1.28 | 0.74 to 2.20 |
| Patient was referred via 2WW pathway. | 22.6 (31/137) | 25.9 (99/382) | 0.84 | 0.42 to 1.66 | 0.98 | 0.55 to 1.74 |
| Patient was referred via emergency pathway. | 0.7 (1/137) | 0.8 (3/382) | 0.93 | 0.10 to 8.94 | * | * |
| Patient was referred via 'other' route. | 1.5 (2/137) | 1.8 (7/382) | | | | |
| Referred patient had an oesophagogastroduodenoscopy (OGD). | 76.8 (53/69) | 75.9 (142/187) | | | | |
| Referred patient was diagnosed with O-G cancer. | 2.9 (2/69) | 3.2 (6/188) | 0.91 | 0.17 to 4.73 | 0.94 | 0.17 to 5.30 |
| Patient referred via standard route was diagnosed with O-G cancer. | 5.7 (2/35) | 1.4 (1/73) | | | | |
| Patient referred via 2WW was diagnosed with O-G cancer. | 0 (0/31) | 5 (5/98) | * | * | * | * |
| Patient referred via emergency pathway was diagnosed with O-G cancer. | 0 (0/1) | 0 (0/3) | | | | |
| Patient was diagnosed with O-G cancer. | 1.5 (2/136) | 2.1 (8/390) | 0.71 | 0.15 to 3.43 | 0.86 | 0.18 to 4.13 |
| Patient diagnosed with O-G cancer had been referred. | 100 (2/2) | 75 (6/8) | * | * | * | * |
| Patient diagnosed with O-G cancer had been referred via standard or 2WW pathway. | 100 (2/2) | 75 (6/8) | * | * | * | * |

Model adjusted for practice size and region; referral status not known—four patients (one intervention and three controls); referral pathway not known—seven control patients; OGD status not known—three patients (one intervention and two controls); O-G cancer status not known—four patients (two interventions and two controls).
*Too few observations to fit the logistic regression model.
n, numerator; N, denominator; O-G, oesophago-gastric; OGD, oesophagogastroduodenoscopy; 2WW, 2-week wait.

**Table 4** Practice and GP demographics

| Practice | | | | GP | | | | | | Interviews | | Claimed use of eCDS |
| Region ID | Size | GPs (n) | Setting | GP ID | Gender | Years registered | Status | Research lead | Sessions/ week | n (months from set-up) | | First follow-up Second/third follow-up |
|---|---|---|---|---|---|---|---|---|---|---|---|---|
| East Practice 1 | 5000–9999 | 7 | Urban | GP3_D | Female | <5 | Registrar | No | 4–6 | 2 (4:8) | | A little bit Once |
| | | | | GP6_D | Male | 20–25 | Partner | Yes | 4–6 | 1 (18) | | Once or twice |
| Northeast Practice 2 | 1000–4999 | 2 | Rural | GP5_D | Female | 10–15 | Partner | Yes | <3 | 2 (19:25) | | Not used Couple of times |
| Northeast Practice 3 | 10 000+ | 5 | Rural | GP8_D | Female | 26–30 | Partner and trainer | No | 7–8 | 1 (19) | | A little bit |
| | | | | GP9_D | Female | 26–30 | Partner and trainer | No | 4–6 | 1 (19) | | Not used |
| East Practice 4 | 10 000+ | 4 | Urban | GP2_E | Male | 10–15 | Partner | No | 7–8 | 1 (3) | | Not known |
| | | | | GP7_E | Female | 20–25 | Partner | Yes | 7–8 | 1 (16) | | Used initially, until NICE NG12 released |
| East Practice 5 | 1000–4999 | 2 | Urban | GP1_D | Female | 20–25 | Partner and trainer | Yes | 10 | 3 (2:6:11) | | Three to four Not used/a few |
| East Practice 6 | 1000–4999 | 2 | Urban | GP4_E | Male | 10–15 | Partner | Yes | <3 | 2 (7:15) | | Two or three Quite often |

eCDS, electronic clinical decision support; GP, general practitioner; NICE, National Institute for Clinical Excellence.

**Table 5** Problems encountered by GPs in use of eCDS

| Problems with access and use of eCDS | |
|---|---|
| Lack of integration of the software with the clinical systems (n=7) | 'Yes, we had plenty of training. The tool itself wasn't difficult to use it's just that it didn't integrate particularly well with our system'. (GP1)<br>'It didn't really integrate very well with SystmOne – you opened it parallel to SystmOne'. (GP8) |
| Slow to access and/or use (n=6) | 'You had to open up something completely separate to the clinical system that you're working in, and when you've got very very limited time that was a negative almost pushing you to not using it'. (GP9)<br>'I wasn't very successful with it to be honest because I found that it slowed the computer down, I had the perception that it slowed the computer down…'. (GP8) |
| Software not compatible and crashes SystmOne (n=4) | 'We've been having quite a lot of issues with it crashing our SystmOne and making everything run very slow'. (GP2) |
| Did not autopopulate (n=3) | 'At the moment, I'm having an issue that the platelets and the demographics are not being automatically populated. I suspect that's just that we've got a version out of date. A couple of weeks back, I did ask the manager just to make sure we got the most recent version'. (GP6) |
| Tool is clunky or confusing to use (n=2) | 'But yes,…it is a little bit clunky because it's not all that obvious that you have to press on "Tools" when you get on to it. And then you get to the "Cancer Decision Support" icon and then you need to pick the right one, so because we don't do it sort of every day or every week, it could be made slightly easier, I think…It's also a little bit confusing that it asks you for a password but you can actually ignore that, but it doesn't feel very logical, you need to have been talked through it once because otherwise it's difficult to figure it out'. (GP1) |
| Problems integrating use of the tool with clinical practice | |
| Not enough time within consultations (n=5) | 'They [patients] never come in with one symptom, or one, sort of, issue, so they come with a few different things, and whether it's psychological or not, the tool really, for my practice anyway, hasn't become embedded …. we won't automatically think, when a patient, like out of three problems, one of them is related to a gastric or oesophageal cancer, erm, I'm not necessarily going into the tool'. (GP5)<br>'No way on this planet any of the GPs under the pressure we were under(…)was going to use a separate program'. (GP8) |
| Did not aid decision making (n=3) | 'So, I put the symptoms in, erm, it just felt, and I documented it in the notes a couple of times I think, but I can't, I couldn't see what it added- I know it's for research, but I couldn't see that it added anything for us, it didn't help me really with any decision-making'. (GP9) |
| Concerns about the accuracy of the data used within the tool (n=4) | 'I feel a bit uncomfortable that the tool requires or populates several boxes with old information'. (GP1)<br>'It had so many different words for very slightly different symptoms and I found that a little bit confusing and I'm not sure that anyone would be… how specific everyone would be about exactly what kind of symptoms the patient had and also if the patient could be particularly specific'. (GP3)<br>'We do a lot of our work by free text. We put under headings in free text. So a lot of symptoms it uses, it won't pick up because it will be in free text. Sometimes it will be there and it will pick up things like the platelets, which is great, and the main thing is it's a gastro-intestinal thing. All the other things that we might put in free text, it wouldn't pick up'. (GP7) |

eCDS, electronic clinical decision support; GP, general practitioner.

So I think before the new cancer guidelines, I thought "Oh yes, that would be good" but since the new cancer guidelines, trying to get my head around those, most of it, what we have at hospital now, we have, probably you know, we have two-week wait proformas. So when we're worried about someone, we tend to just look on the pro forma to see where they, that's how I work really. (GP7)

On the positive side, many of the GPs welcomed the prospect of a tool that could help to communicate risk to patients and provide them with clinical grounds for referral rather than a clinical 'hunch'. They particularly saw the value of having a tool to use with anxious patients who were at low risk, with six GPs thinking they would be most likely to use any eCDS tool with patients who were overanxious or worried about their cancer risk when they themselves saw little reason for concern:

It can be used in a consultation, that's where it comes in handy, just to reassure a patient when my gut instinct is not to be too worried. (GP4).

I mean if you've got a patient who is sat there and has come in saying "I think I have got, oesophageal cancer" or something, then you're going to be taking that consultation from a completely different tack, you then take the history, you go- you know rationalise everything with the patient in terms of what puts them at risk, what doesn't put them at risk and then using a tool in that circumstance to definitely show them that, numerically, you know their risk is low. (GP9)

Three GPs thought the main reason for using any eCDS tool was to achieve patient benefit: 'So I think if you've got a purpose for it and it makes sense to you that it's something that actually will help you look after your patients better, we'll always try to use it' (GP8). There was no record of these GPs having used the study eCDS tool.

### Health economic analysis

This analysis predicts the eCDS tool to save 0.028 quality-adjusted life years (QALYs) (95% CI −0.014 to 0.071) and 0.008 life years (95% CI −0.014 to 0.035) per person consulting a GP with symptoms. These benefits come primarily from reducing the number of emergency referrals—the eCDS is projected to prevent 17 (95% CI 3 to 216) emergency referrals per 10 000 individuals consulting the GP with symptoms. The maximum cost that eCDS installation and training could be and still enable the intervention to be cost-effective was estimated assuming a willingness to pay threshold of £20 000 per QALY. The maximum cost per person consulting the GP with symptoms was £569 (95% CI −£265 to £1402), but for the eCDS to save costs in the long run, this reduces to £6 (95% CI −£29 to £53). However, the 95% credible intervals indicate high uncertainty, with a 9.7% probability that eCDS produces a QALY loss and an 8.8% probability that any cost at all for eCDS installation and training would be too high to enable it to be cost-effective at the £20 000 willingness to pay threshold. A complete report of the health economic analysis is available online.[18]

### DISCUSSION

In this feasibility study of eCDS in primary care for detecting possible O-G cancer, we found that GPs used the tool very infrequently and that poor integration of the eCDS tool with the GP workflow was an evident problem. Implementation of eCDS in intervention practices was seriously disrupted by technical, regulatory and organisational obstacles that emerged only at the point of installation on practice computer systems. Any definitive trial of eCDS for cancer diagnosis that has clinical endpoints will likely require a very large number of participating practices for adequate power. It is possible that eCDS for suspected O-G cancer in primary care could be cost-effective with lower implementation costs, but the data generated by this study were insufficient to support such a recommendation.

The strengths of this study included its use of a previously piloted eCDS tool and a theory-based approach to implementation. It was a pragmatic study, with GPs in the intervention practices free to use the tool as and when they thought it necessary, reflecting the way that eCDS would be used in daily practice. It addressed a problem of identifying patients with upper gastrointestinal symptoms who require further evaluation, which is common in primary care and places substantial demand on specialist services. We successfully optimised the intervention software to enable data capture for the purpose of research. We obtained valuable insights to inform the design and conduct of a definitive trial.[19] There were, however, several weaknesses. First, two-thirds of patients identified as potentially eligible proved not to be so on scrutiny of the clinical records. This was despite careful and iterative development of the search strategy with the North Tees PCT Information and Computing Service to minimise errors of inclusion. Two study clinicians (GR and FW) reviewed the screening process with several practices but failed to identify any systematic errors giving rise to unwarranted exclusion. While undercoding of diagnoses may have reduced the number of eligible patients, any consequent prescription should have been identified. Second, recruitment of patients to the study was lower than anticipated, at 33% of those invited. Third, the eCDS tool interfaced poorly with the SystmOne clinical software, a problem not reported in the preceding Macmillan NAEDI pilot. This made it slow to use and the software developers and TPP were unable to identify a remedy. Fourth, new restrictions on the uploading of software to GP clinical systems were introduced by PCTs early in the study. The way in which these were applied in one study region resulted in long delays in activating the intervention arm of the study. Fifth, the introduction of revised NICE guidelines for management of suspected cancer early in the recruitment period was perceived by some GPs to supersede the need for an eCDS tool. The poor integration of the eCDS tool with the GP workflow and rarely perceived need for its use also impacted on recruitment of GPs for interview. Lastly, the small sample size of the study data, in particular the extremely small numbers diagnosed with O-G cancer, resulted in high uncertainty in the health economic model.

There are several published reports of eCDS tools for cancer diagnosis in primary care.[13] Of these, only one has been an RCT of an eCDS tool designed to support the GP's assessment at the time of consultation of the risk of suspected cancer and to inform their decision on whether to refer for specialist assessment. That trial showed that a system that integrated a primary care algorithm for suspected melanoma and SIAscopy (MoleMate) did not improve case selection for referral compared with standardised use of the Seven Point Checklist, due to the low specificity of the diagnostic algorithm.[20] Of the remaining reports, one was of a laboratory-generated standard text prompt for clinical management of patients with a full blood count consistent with iron deficiency anaemia,[21]

while two others were of computer algorithms to retrospectively identify red flag features in the clinical records and flag the record for follow-up or further action.[22 23] A fourth study was of a computer-based referral template intended to improve the information contained in referrals letters.[24] Of these, only one demonstrated a significant effect, shortening the time to diagnostic evaluation for patients with colorectal and prostate, but not lung, cancers.[23]

Use of the eCDS tool in this study was disappointingly low. We used an implementation approach, educational outreach, which is well established and theoretically based. However, interventions to change professional behaviour have effect sizes that are modest at best.[17] In order to avoid the well-recognised problem of prompt fatigue, we chose not to include this feature in our eCDS. Prompts and the requirement for practitioners to justify over-riding them have, however, been identified as one of the few features of eCDS associated with improved process of care.[12] The most recent systematic review of eCDS for processes of care draws attention to the complex sociotechnical context in which eCDS is used, reports only a small to moderate improvement in targeted processes of care and concludes that the predictors of meaningful improvement remain undefined.[25] Evaluations of eCDS for suspected cancer should specifically address the sociotechnical context of their use. Tools such as the safety-related electronic health record research reporting framework (SAFER),[26] developed specifically to address the multidimensional nature of such interventions, should be considered for this purpose.

We identified that this eCDS tool had a role in supporting communication around patient care decisions, particularly for anxious patients considered at low risk by the GPs. However, we also found evidence to support the three core constructs related to use of these tools that have been described by others: trust; the GP's role as a gatekeeper; and the impact on workflow.[13] Specifically, GPs' accounts reflected how they did not always trust the data used to populate the tool, how difficult it was to commit to working with a tool that was not integrated into their operating system and how the tool appeared to slow their computer processes down. The importance of integration of tools for GPs was also a key finding in an evaluation of an eCDS tool for melanoma.[27]

Only one trial of eCDS for suspected cancer in primary care has been the subject of a formal health economic analysis. The MoleMate eCDS tool was considered to be cost-effective, with an incremental cost-effectiveness ratio (ICER) of £1896 per QALY gained, but with considerable decision uncertainty related to the sensitivity and specificity of MoleMate when compared with best practice.[28] We consider it possible that eCDS for suspected O-G cancer in primary care could be cost-effective if implementation costs are minimised.

A key finding from this study is how highly susceptible implementation of eCDS in primary care is to technical and organisational considerations. These include the quality of the interface between the eCDS tool and the clinical system and the ease of use, one with the other. Furthermore, use of eCDS in clinical practice is sensitive to how well it integrates with the GP workflow and the frequency with which users perceive a need for it. These factors will also be relevant to the introduction of eCDS in other national healthcare systems. However, some challenges specific to the English healthcare system were apparent. We found the installation of the research software on practice computer systems became subject to regulatory controls during the implementation phase of the study, and that these differed between PCTs, attracted a significant charge in one case and changed over time. These administrative restrictions could not have been foreseen but seriously disrupted the smooth running of the study. Any definitive trial of eCDS for cancer diagnosis should not be done without further development of the intervention to address the limitations we describe.

In conclusion, to be of practical use in the consultation, an eCDS tool for suspected cancer in primary care should be technically well integrated with the clinical software used by the GP, easily accessed from within that system and not impact on its operation. Even then, it is likely to be used infrequently and any pragmatic trial of its impact on clinical outcomes should be powered accordingly.

**Author affiliations**
¹Institute of Population Health Sciences, University of Newcastle upon Tyne, Newcastle upon Tyne, UK
²Department of Public Health and Primary Care, University of Cambridge, Cambridge, UK
³Department of General Practice and Centre for Cancer Research, University of Melbourne, Melbourne, Victoria, Australia
⁴Primary Care Diagnostics, University of Exeter Medical School, Exeter, UK
⁵North Wales Organisation for Randomised Trials in Health, Bangor University, Bangor, UK
⁶School of Health and Life Sciences, Teesside University, Middlesbrough, UK
⁷School of Health and Related Research (ScHARR), University of Sheffield, Sheffield, UK
⁸NIHR CLAHRC South West Peninsula (PenCLAHRC), University of Exeter Medical School, Exeter, UK
⁹School of Medicine, University of Leeds, Leeds, UK

**Acknowledgements** The authors thank Anisah Tariq, Helen Moore, Christina Dobson, Andy Cowan, Fiona Scheibl, Nicola Hall, Anne Kershenbaum and Anna Wood for contributions to the conduct of this research; North Tees Information and Computing Services for development of the search strategy; Primary Care Research Networkss and practices in NE and Cumbria, Yorkshire and East of England for their participation; TPP for modification of the electronic clinical decision support tool and data capture, and for use of their clinical tools platform; the Trial Steering Committee, chaired by Professor Chris Salisbury.

**Contributors** GR, FMW and RDN conceived the research and led the study. JE, WH, ZH, OCU, SW and JAU-S contributed to the design. CN, JH and AW contributed to the conduct of the study. ZH provided Clinical Trials Unit support. SW, CT and TS were responsible for the health economic evaluation. OCU conducted the statistical analysis. GR drafted the manuscript with contributions from FMW, RDN, TS and JAU-S. All authors reviewed the manuscript and approved the final version.

**Funding** This study was supported by Cancer Research UK (reference number c6971/A17940) and the Policy Research Unit for Cancer Awareness, Screening and Early Diagnosis, Policy Research Programme 106/0001. This research arises from the CanTest Collaborative, which is funded by Cancer Research UK (C8640/

A23385), of which FMW and WH are directors and GR, JE and RDN are associate directors. Sponsor: Durham University (protocol V.4.0 dated 13 July 2015).

**Competing interests** JAU-S was funded by a CRUK Prevention Fellowship (C55650/A21464). OCU was supported by the National Institute for Health Research Applied Research Collaboration South West Peninsula.

**Patient consent for publication** Not required.

**Ethics approval** This study received ethics approval from the NHS Health Research Authority National Research Ethics Service (14/NE/1179)

**Provenance and peer review** Not commissioned; externally peer reviewed.

**Data availability statement** Anonymised participant data are available upon reasonable request from bona fide researchers. Please contact Gregory.rubin@ncl.ac.uk or fmw22@medschl.cam.ac.uk.

**ORCID iDs**
Greg Rubin http://orcid.org/0000-0002-4967-0297
Fiona M Walter http://orcid.org/0000-0002-7191-6476
Jon Emery http://orcid.org/0000-0002-5274-6336
Willie Hamilton http://orcid.org/0000-0003-1611-1373
Zoe Hoare http://orcid.org/0000-0003-1803-5482
Chloe Thomas http://orcid.org/0000-0001-8704-3262
Obioha C Ukoumunne http://orcid.org/0000-0002-0551-9157
Juliet A Usher-Smith http://orcid.org/0000-0002-8501-2531
Richard D Neal http://orcid.org/0000-0002-3544-2744

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
