## [Reviewer comments · BMJ Open]

ARTICLE DETAILS

TITLE (PROVISIONAL)	An electronic clinical decision support tool for assessing stomach symptoms in primary care (ECASS): a feasibility study
AUTHORS	Rubin, Greg; Walter, Fiona; Emery, Jon; Hamilton, Willie; Hoare, Zoe; Howse, Jenny; Nixon, Catherine; Srivastava, Tushar; Thomas, Chloe; Ukoumunne, Obioha; Usher-Smith, Juliet; Whyte, Sophie; Neal, Richard

VERSION 1 – REVIEW

REVIEWER	Prof David Bowrey Dept of Surgery University Hospitals of Leicester NHS Trust Leicester UK
REVIEW RETURNED	04-Aug-2020

GENERAL COMMENTS	This is a well written, interesting manuscript that addresses a potentially important question. The trial findings are disappointing, with uptake among GPs poor and a consequent low recruitment rate of around 10% of eligible participants. 76% of patients in both groups underwent endoscopy. In this patient cohort, no cost savings or better cancer pickup rate were identified. This makes meaningful interpretation limited. The authors should highlight the weaknesses in the trial conduct more strongly than they have. The take home message should be that the electronic decision tool should be scrapped or a robust educational process put in place, before any further money is spent upscaling this project. From a funding point of view, continuity funding should only be given, on the basis of addressing these methodological flaws in trial conduct. They allude to the problem with the paragraph, " The eCDS tool could be accessed and the output utilised at the GP's discretion. As configured for this study, it did not generate 'prompts'. The installation of software on practice computer systems for the purpose of research became subject to new regulatory controls during the implementation phase of the trial. These differed between Primary Care Trusts and changed over the course of the study, disrupting the smooth operation of the intervention arm of the trial". It was disappointing that a more defined GP education intervention was not put in place before the trial started. I am concerned that the study was destined to "fail" before it started because of this. There did not seem to be buy-in from GPs and a general reluctance to use the electronic decision making tool. They should discuss what educational measures should be used to get over this hurdle
---

VERSION 1 – AUTHOR RESPONSE

Reviewer 1	
This is a well written, interesting manuscript that addresses a potentially important question. The trial findings are disappointing, with uptake among GPs poor and a consequent low recruitment rate of around 10% of eligible participants. 76% of patients in both groups underwent endoscopy. In this patient cohort, no cost savings or better cancer pickup rate were identified. This makes meaningful interpretation limited.	Thank you for these comments. We agree that the uptake and recruitment rates were low, but in our view confirm the value of our feasibility study. These were important findings that will inform, indeed have informed, the design and power of current definitive trials such as ERICA www.theericatrial.co.uk ISRCTN 22560297.
The authors should highlight the weaknesses in the trial conduct more strongly than they have. The take home message should be that the electronic decision tool should be scrapped or a robust educational process put in place, before any further money is spent upscaling this project. From a funding point of view, continuity funding should only be given, on the basis of addressing these methodological flaws in trial conduct.	We have considerably extended the paragraph on strengths and weaknesses of the study. This was a feasibility study and not a definitive assessment of a health technology. As such, it would not be appropriate to use our findings to decide the fate of an eCDS tool. We believe our use of educational outreach was a well-recognised, theoretically and evidence-based approach to implementation. We have described the educational package in more detail to help the reader in the methods section. In the discussion we now argue that implementation of eCDS has more recently come to be understood as a complex socio-technical process and that future evaluations should take account of this. We would contend that the study was carefully designed and theory-based, but we completely agree that any future research funding should be contingent on the design of the proposed trial addressing the methodological problems we encountered when it came to running our study. We make this point strongly in our discussion.
It was disappointing that a more defined GP education intervention was not put in place before the trial started. I am concerned that the study was destined to "fail" before it started because of this. There did not seem to be buy-in from GPs and a general reluctance to use the electronic decision-making tool. They should discuss what educational measures should be used to get over this hurdle	We thank the reviewer for these comments, and in response have provided a more detailed description in the methods section of the implementation package that was provided to intervention practices. This package was based on educational outreach, a well-established approach to effecting professional behaviour change. The effectiveness this and other approaches is the subject of a Cochrane review, which we used and now reference. Unfortunately, all interventions to effect professional behaviour change have effect sizes that are modest at best. We have, however, extended the discussion to consider how more recent systematic review insights can inform the design of eCDS tools, and how research into their implementation should take account of the complex socio-technical context in which they are used.